# OpenReview forum: "Turning Bias into Bugs: Bandit-Guided Style Manipulation Attacks on LLM Judges"
_ICML.cc/2026/Conference — ICML 2026 regular_

### Official Review · Reviewer_BEoC · 2026-03-06

**Soundness:** 2
**Presentation:** 3
**Significance:** 3
**Originality:** 2
**Overall Recommendation:** 2
**Confidence:** 4

**Summary:**

BITE is a black-box adversarial framework that learns semantics-preserving edits to mislead the judgment and artificially inflate judged scores. Authors cast the selection of stylistic edits as a contextual bandit problem and use a LinUCB policy to adaptively choose edits that maximize the judge’s score without access to model parameters or gradients.

**Compliance With Llm Reviewing Policy:**

Affirmed.

**Final Justification:**

This paper proposed a black-box adversarial framework that learns semantics-preserving edits to mislead the judgment and artificially inflate judged scores, namely BITE. However, BITE heavy reliance on multi-round online queries (25 iterations in the paper), and it completely fails to execute in many real-world application scenarios (e.g., zero-shot blind evaluations). Consequently, I feel that significant refinements are needed to improve the practical application of this work.

**Key Questions For Authors:**

1. A review of Table 6 reveals a very high attack success rate for BITE on objective questions in the AlpacaEval dataset under the pointwise setting. However, the success rate drops significantly under the pairwise setting. The pairwise approach is closer to the industry RLHF standard. The ArenaHard dataset exhibits the exact same pattern. Why does pairwise comparison provide much stronger natural resistance to this stylistic attack? In a pairwise setting, the judge evaluates and compares two answers simultaneously. The paper lacks an in-depth discussion on this phenomenon.

2. Given that the exploration boundary of the BITE framework is confined to the eight predefined stylistic features, would the effectiveness of the framework be significantly reduced if the target model does not have a preference for these eight styles?

**Limitations:**

Yes.

**Strengths And Weaknesses:**

Strengths:
Presentation: The paper is well-organized and easy to understand. It features a clear and logical structure.
Significance: The authors raise an interesting question. They explore whether stylistic biases in current large language models create security vulnerabilities. Furthermore, the authors evaluate the stealthiness of attacks that use this vulnerability to manipulate agent evaluations.

Weaknesses:
1. The authors claim to provide a rigorous regret bound proof for LinUCB under model misspecification. However, the core derivation heavily relies on classical works regarding self-normalization bounds. The authors' approach to handling nonlinear bias is theoretically quite conventional.
2. An attack strategy optimized for a specific judge model is difficult to transfer to another. The attack success rates in areas outside the main diagonal are very low, as shown in the heatmap in Figure 4. This implies that the success of BITE highly depends on repeated API queries to the same evaluation judge. In actual evaluation tasks, judge models are often hidden, dynamically updated, or integrated into an ensemble. The effectiveness of BITE will likely drop significantly if the attacker cannot probe the target judge for the required number of rounds before submission.
3. In Table 13, the rewriting attack reduces the text score after a BITE attack. The score drops from 8.55 to 6.67, which is close to the initial score of 6.35. The authors explain that the rewriting attack is not a reliable universal defense. However, it is undeniably a widely used method. The authors' explanation still cannot conceal the fact that the BITE attack cannot survive under rewriting.
4. The use of the all-MiniLM-L6-v2 model is crucial to the BITE framework. If this embedding model cannot effectively capture the required stylistic features, the agent is likely exploring blindly in a very vague feature space. The all-MiniLM-L6-v2 model is specifically trained to capture semantic similarity. Therefore, it is insensitive to surface features like style, tone, and formatting structure. The authors use a model designed for semantics to guide the agent in choosing stylistic modifications. I believe this creates a logical mismatch.
5. The authors assume that future conferences might adopt a centralized review system (in Section 6.2.2). This would allow attackers to interact with the same judge across multiple venues or submission cycles. However, the real academic submission process is quite different. An author might submit 25 slightly modified versions of the same paper just to probe the judge's preference. These 25 versions would only differ in style, formatting, and emojis. Doing this would very likely trigger plagiarism checks or anti spam mechanisms. The authors do not discuss the stealthiness of such high frequency probing with highly similar texts. Consequently, I believe that the practical applicability of the BITE framework in real-world scenarios is significantly limited.
6. There are multiple notation errors in the proof in Appendix C. I suggest the authors review and correct them in subsequent revisions.

---

> ### Author Rebuttal · Authors · 2026-03-29
>
> **W1: Theoretical proof is conventional.** We clarify that our core contribution is the formulation of a specialized theoretical framework for BITE attacks in LLMs, adapting the misspecified linear bandit model for black-box interactions. Our work addresses a more complex scenario than that covered by standard LinUCB. The model's unique multi-arm, multi-parameter structure—where each action has its own parameter $\theta_b$ and action-dependent misspecification—renders existing theoretical results inapplicable. Thus, the novelty lies not in creating new fundamental proof techniques, but in the non-trivial adaptation and extension of bandit theory required to analyze this new and important problem setting.
>
> ---
> **W2: Clarification on Threat Model & Transferability.** BITE does not require *offline* probing to target model before evaluation; our scope is a practical and widely-used threat model: repeated black-box interaction with a stable or moderately evolving evaluation endpoint. Under this black-box formulation, the hidden judges and fixed ensembles mentioned remain fully in-scope. Highly dynamically updated evaluators are stronger defense settings and are beyond our current scope. Crucially, the low off-diagonal transferability is not a limitation but our core motivation: because stylistic vulnerabilities are heavily evaluator-specific, static pre-computed attacks are unviable. This necessitates an adaptive, online attack exactly like BITE.
>
> ---
> **W3: Clarification on rewrite defense.** We agree rewriting can be a strong mitigation, but is not a default defense for LLM judges. Universal rewriting is heavily task-dependent: it risks distorting normal answers, incurs substantial token/computational costs, and suppresses legitimate stylistic signals. Therefore, our claim is that BITE remains a highly relevant threat in domains where rewriting every input is inappropriate, such as paper reviewing, benchmarking, and data curation. We will make this clearer in the appendix.
>
> ---
> **W4: Style Sensitivity of the Embedding of all-MiniLM-L6-v2.** While all-MiniLM-L6-v2 is semantics-oriented during training, empirical data proves it is not blind to stylistic surface features. Across 17,521 pairs of original vs. style-modified answers, the average cosine similarity drops to 0.922. Therefore, MiniLM does not collapse stylistic variations into identical representations.
>
> Crucially, BITE's contextual bandit does not require a perfectly precise style encoder. The algorithm is inherently robust to coarse or noisy state spaces; as long as the representations are distinguishable (which our data proves), the agent successfully leverages reward signals to optimize modifications, rather than "exploring blindly."
>
> ---
> **W5: Practical Applicability concern of BITE in paper reviewing.** To clarify, our threat model is not about submitting 25 near-duplicate papers at once. BITE is a general black-box attack under a limited *sequential* query budget.
> The paper-reviewing section is a ***forward-looking case study*** of potential vulnerabilities in future centralized review systems. The 25-query budget is an experimental upper bound, not a practical requirement, as gains are significant in early rounds (5-10 round as shown in Fig.2). While operational defenses like spam detection are vital, they are complementary. Our core finding holds: LLM judges can be manipulated via stylistic edits under tight budgets.
>
> ---
> **W6: Notation errors.** We will correct the notation errors in the revision.
>
> ---
> **Q1: Why is ASR higher in pointwise than pairwise evaluation?** We hypothesize that the gap mainly comes from the stricter success criterion in pairwise evaluation. In pointwise evaluation, the attack only needs to increase the score of the attacked answer itself. In pairwise evaluation, it must increase the attacked answer enough to outperform the reference. Hence, many attacks that successfully inflate absolute scores do not necessarily flip the relative comparison. This suggests that stylistic bias is easier to exploit in absolute scoring than in comparative judging. Our takeaway is therefore not that pairwise evaluation is robust, but that pointwise judging is especially vulnerable.
>
> ---
> **Q2: Would BITE weaken if the judge is insensitive to the 8 predefined styles?** Our current implementation would be less effective if a judge were insensitive to the chosen stylistic actions. However, this is unlikely in practice, as stylistic bias is a persistent and well-documented issue in LLM judges, and our eight transformations are well-grounded in existing literature. More importantly, our main contribution is the **BITE framework**, not this fixed bias set. BITE is a black-box adaptive method for exploiting stylistic vulnerabilities, and the action space can be extended as new biases are identified. In this sense, bias discovery and BITE are complementary: one finds vulnerabilities, while BITE provides an efficient mechanism to exploit them.

---

> > ### Author Rebuttal · Reviewer_BEoC · 2026-04-02
> >
> > Thanks for the response. I have carefully read the authors' rebuttal, which addressed some of my concerns. However, the authors overstate the implications of Figure 4. The results merely prove that overfitted adversarial samples generalize poorly, not that all static attacks are categorically invalid. This logical leap effectively masks BITE's critical weakness: its heavy reliance on multi-round online queries (25 iterations).
> >
> > In many high-stakes adversarial scenarios (e.g., zero-shot blind evaluations), attackers are denied the privilege of iterative probing. Under such constraints, BITE does not merely suffer from low transferability, it completely fails to execute. Given these severe practical limitations, I have decided to maintain my score.

---

> > > ### Author Response · Authors · 2026-04-04
> > >
> > > We thank the reviewer for their follow-up questions. Given that this final reply focuses exclusively on our wording regarding static attacks and the pratical limitation of our multi-round threat model, we hope to resolve these remaining concerns.
> > >
> > > ---
> > > **Clarification on Static Attacks**: Regarding the overstated sentence, we clarify our position accordingly: because precomputed stylistic biases transfer poorly, an adaptive approach like BITE remains necessary to reliably exploit them. This low transferability is not a limitation of the BITE framework, but the inherent nature of stylistic biases across different contexts. By utilizing an adaptive approach, BITE successfully overcomes this barrier to effectively exploit these biases.
> > >
> > > ---
> > > **Feasibility of the Multi-Round Threat Model**: Regarding the reviewer's concern about *our reliance on multi-round queries (25 iterations)*, we respectfully emphasize that iterative probing **is not an unstated limitation, but our explicitly defined threat model** (Sections 3.2 and 4). We further clarify the feasibility and urgency of this threat model as follows:
> > >
> > > 1. **This threat model covers highly practical and widely deployed scenarios.** As explicitly motivated in the paper (Lines 33-37, 112-142), we specifically target prevalent judge-based deployments where adaptive interaction is common practice. For example, in leaderboards (like Kaggle or Hugging Face Open LLM Leaderboard), users can exploit iterative submission allowances to optimize for judge biases, such as verbosity, rather than true performance. Similarly, RLHF training inherently enables "reward hacking," where policy models learn to exploit reward model vulnerabilities over millions of steps. Finally, adversaries can use free moderation APIs (e.g., OpenAI’s Moderation API) for iterative "pre-flight" probing to reliably bypass data curation filters. These scenarios illustrate a clear shift from one-off attacks to sustained, optimization-driven exploitation.
> > >
> > >
> > > 2. The adaptive attack **is firmly established and well-acknowledged by the research community**. In [2], top AI security researchers including Nicholas Carlini argue that "success on static evaluations provides only a false sense of security", and "we should evaluate defenses against adaptive attackers who explicitly modify their attack strategy to counter a defense’s design while spending considerable resources to optimize their objective." Additionally, recent top-tier security literature JudgeDeceiver [1] recognizes LLM-as-a-Judge as a critical attack surface susceptible to iterative manipulation in RLAIF settings. Therefore, this threat model is viewed by the community as both an urgent necessity and a practical baseline for security evaluations.
> > >
> > >
> > > 3. **Efficiency of the Query Budget**: BITE's 25-query budget is highly conservative compared to standard black-box attacks (e.g., ~1000 queries for GCG [3], ~600 for JudgeDeceiver [1], ~100 for AutoDAN [4], and 15–100 for TAP/PAIR [5, 6]). This negligible overhead allows for easy execution in practical settings like automated grading. Importantly, 25 queries is only an experimental upper bound as clearly stated in our previous rebuttal, BITE already achieves substantial score inflation within just 5–10 rounds (Fig. 2), underscoring its immediate threat.
> > >
> > >
> > > Therefore, `we respectfully but strongly disagree that our method's importance is negated because "In many high-stakes adversarial scenarios (e.g., zero-shot blind evaluations), attackers are denied the privilege of iterative probing."` Points 1-3 above establish that adaptive attacks like ours are an urgent concern in the community, directly applicable to many practical settings, and our work matches the threat models in many established works. The reviewer's objection would directly apply to all adaptive attacks, suggesting that they are not valuable. However, a contribution is valuable if it covers sufficiently broad and critical practical settings; it would not be reasonable to expect it to apply to every scenario, particularly those not in the scope it is designed for. Accordingly, our abstract and introduction clearly state our scope as the adaptive attack scenario, and our threat model section formalizes this.
> > >
> > >
> > > [1] Optimization-based prompt injection attack to llm-as-a-judge.
> > >
> > > [2] The attacker moves second: Stronger adaptive attacks bypass defenses against LLM jailbreaks and prompt injections.
> > >
> > > [3] Universal and transferable adversarial attacks on aligned language models.
> > >
> > > [4] AutoDAN: Generating Stealthy Jailbreak Prompts on Aligned Large Language Models.
> > >
> > > [5] Tree of attacks: Jailbreaking black-box llms automatically.
> > >
> > > [6] Jailbreaking black box large language models in twenty queries.

---

### Official Review · Reviewer_ELhL · 2026-03-09

**Soundness:** 3
**Presentation:** 3
**Significance:** 3
**Originality:** 3
**Overall Recommendation:** 4
**Confidence:** 3

**Summary:**

This paper turns stylistic bias in LLM judges from a static weakness into an attack surface that can be actively explored and exploited. It proposes BITE, a contextual bandit framework that iteratively applies semantics-preserving style manipulations to candidate answers and uses judge score improvements as reward signals, with the goal of finding answer styles that systematically obtain higher scores from black-box judges. The paper evaluates the method across pointwise and pairwise benchmarks, multiple judge models, and an automated peer-review case study, and further provides a regret-style analysis for a misspecified linear bandit formulation. Overall, the paper raises an important and timely concern: LLM-based evaluation can be manipulated not only by changing content, but also by strategically optimizing presentation style.

**Compliance With Llm Reviewing Policy:**

Affirmed.

**Final Justification:**

The authors' response resolves my previous concerns to some extent. I think the paper sound, clear, and somewhat innovative. I choose to maintain my current score.

**Key Questions For Authors:**

1. The reward used in the theoretical analysis does not appear fully aligned with the reward used by the actual algorithm. In the method section, LinUCB is updated using the marginal score improvement $r_t$, where $r_t = S_t - S_{t-1}$. However, the theory models the observed reward as $y_t = x_t^\top \theta^*_{b_t} + \eta_t + m_t(b_t)$, which is a standard contextual bandit payoff depending on the current context-action pair. The issue is that $r_t$ actually depends on the parent node's previous score $S_{t-1}$, rather than being a simple payoff determined only by the current $(x_t, b_t)$. In other words, the practical process seems closer to a path-dependent or parent-dependent non-stationary reward model, whereas the theory simplifies it into a standard contextual bandit observation model. Could the authors clarify this gap and explain why the current theoretical abstraction is appropriate?

2. The misspecification level $\zeta_T$ is assumed to be known to the algorithm, which seems rather strong in a realistic black-box setting. How should practitioners interpret this assumption, and is there any practical way to estimate or upper-bound $\zeta_T$ in the attack scenario considered in the paper?

3. For the results in Table 2, what is the score of the original unattacked initial reviews before any rewriting or attack is applied? The table reports final review scores for several methods, but without the initial baseline score it is difficult to assess the absolute gain introduced by each method.

4. Could the authors clarify what each x-axis label in Figure 5 corresponds to, especially whether "UCB" stands for BITE? More broadly, the figure seems to show that style-control defense has little effect on all four attack strategies, not only on UCB. In that case, does Figure 5 mainly demonstrate that this defense is generally weak, rather than that BITE is uniquely difficult to defend against?

**Limitations:**

See Questions and Weaknesses.

**Strengths And Weaknesses:**

### Strengths
- The paper is well-structured and easy to follow.
- Thorough experimental verification has been conducted.
- The theoretical analysis is carefully presented.

### Weaknesses
- Formatting issues, such as the same paper "Optimization-based prompt injection attack to llm-as-a-judge" being cited twice in the references.
- Some experimental details are not clearly demonstrated.

---

> ### Author Rebuttal · Authors · 2026-03-29
>
> **W: Format issues like Duplicate reference \&\& mismatch x-axis label in Figure 5**
>
> Thanks for pointing them out. We will fix them in the revision.
>
> ---
> **Q1: Clarification on Parent-Relative Reward and State Abstraction**
>
> We clarify that while the practical reward is defined as a marginal improvement $r_t = S_t - S_{t-1}$, this does not introduce additional path dependence beyond the current context.
>
> Specifically, our context at time $t$ includes the question-answer pair $(q, a_{t-1})$, and the context vector is constructed as $x_t = \phi(q, a_{t-1})$. The previous score $S_{t-1}$ is obtained from the judge evaluation instance $J(q, a_{t-1})$, and therefore does not act as an additional independent state variable outside the information used to construct the current context. Accordingly, the reward can be rewritten as $r_t = J\ \left(q, \psi(a_{t-1}, b_t)\right) - J(q, a_{t-1}) = f \left((q, a_{t-1}), b_t\right)$, where $b_t$ is the selected stylistic action. Hence, the reward is driven by the current context-action pair, rather than the full trajectory. In this sense, the appearance of $S_{t-1}$ reflects our use of local score improvement relative to the current answer, rather than a separate parent-dependent variable outside the current context representation. We will clarify this point in the revision.
>
> ---
> **Q2: Practical Interpretation of the Misspecification Level $\zeta_T$**
>
> From a practitioner’s perspective, directly estimating or upper-bounding $\zeta_T$ is indeed difficult: it is a horizon-level quantity and can vary across tasks, judges, and interaction regimes. Instead, in the implementation, its effect is **absorbed into the exploration parameter $\alpha$** in LinUCB (see Theorem 5.1). Concretely, we tune $\alpha$ via hyperparameter search rather than estimating $\zeta_T$. Empirically, we observe that a stable range (e.g., $\alpha \approx 1.0$) consistently yields strong performance across different judges and datasets, suggesting that the method is not *highly* sensitive to the exact value.
>
> This aligns with standard practice in bandit algorithms, where theoretical quantities (e.g., [1,2] ) inform the design of exploration but are not directly estimated in deployment. We will clarify this point in the revision.
>
> [1] Li, Chu, Langford, and Schapire (2010), A Contextual-Bandit Approach to Personalized News Article Recommendation.
>
> [2] Abbasi-Yadkori, Pál, and Szepesvári (2011), Improved Algorithms for Linear Stochastic Bandits.
>
> ---
> **Q3: Table 2 initial baseline score missing**
>
> We thank the reviewer for pointing this out. We completely agree that including the original baseline score is crucial to assess the true impact of our attack.
> In our revised manuscript, we will update Table 2 to include the `Initial scores` alongside the attack results, as shown below:
>
> | Judge Model | `Initial Score` | Iterative Rewrite | Random Action | BITE (ours) |
> | :--- | :--- | :--- | :--- | :--- |
> | deepseek-r1-0528 | 5.67 ± 0.97 | 6.84 ± 0.22 | 7.31 ± 0.23 | 7.63 ± 0.29 |
> | gemini-2.5-flash | 5.28 ± 0.66 | 6.59 ± 0.39 | 7.18 ± 0.28 | 7.44 ± 0.40 |
> | llama-3.3-70b-instruct | 7.90 ± 0.07 | 8.17 ± 0.24 | 8.34 ± 0.19 | 8.38 ± 0.18 |
> | o3-mini | 7.43 ± 0.09 | 7.53 ± 0.17 | 7.53 ± 0.09 | 7.67 ± 0.17 |
> | qwen3-235b-a22b-2507 | 6.01 ± 0.19 | 6.37 ± 0.09 | 6.43 ± 0.21 | 6.50 ± 0.22 |
>
> ---
>
> **Q4: Figure 5 Legend Typo**
>
> We thank the reviewer for pointing this out. The ambiguity comes from a plot-labeling mismatch, which we will fix in the revision. In Figure 5, the four x-axis labels correspond to the following methods: Holistic Rewrite = Holistic Rewrite, Holistic Rewrite-M = Iterative Rewrite, Random = Random Action, and UCB = BITE (ours). These are the same four attack methods used throughout the paper.
>
> **Figure 5 Interpretation**. We agree with the reviewer’s interpretation. Figure 5 is not intended to show that BITE is uniquely able to bypass the style-control defense. Rather, the purpose of this figure is to show that this defense has limited effect on the class of semantics-preserving attacks considered in our paper, including BITE.

---

> > ### Author Rebuttal · Reviewer_ELhL · 2026-04-01
> >
> > The authors’ response is clear and addresses my concerns well.

---

### Official Review · Reviewer_1YdF · 2026-03-12

**Soundness:** 3
**Presentation:** 3
**Significance:** 4
**Originality:** 3
**Overall Recommendation:** 5
**Confidence:** 3

**Summary:**

The paper introduces BITE (Blas exploraTion and Exploitation), a black-box adversarial framework designed to exploit the stylistic biases of Large Language Models (LLMs) used as automated judges. Recognizing that LLM evaluators often favor superficial formatting (e.g., verbosity, markdown, specific sentiments) over substantive quality, the authors cast the generation of adversarial, semantic-preserving stylistic edits as a contextual bandit problem. Using a LinUCB policy, the attacker iteratively probes the judge to uncover its unique "vulnerability fingerprint" and maximize score inflation.The work provides a formal regret bound accounting for the non-linear model misspecification inherent in prompting an LLM judge. Empirically, BITE is evaluated across diverse proprietary and open-source models on benchmarks like AlpacaEval 2.0 and Arena-Hard-Auto. The framework achieves an attack success rate of greater than 65%, successfully evades standard style-control defenses, and highlights that different LLMs exhibit distinct, largely non-transferable stylistic biases.

**Compliance With Llm Reviewing Policy:**

Affirmed.

**Final Justification:**

The authors' response was clear. I choose to maintain my positive rating.

**Key Questions For Authors:**

1. How much does this attack cost in reality? For modeling budget restriction, you have set $T=25$, but stating the actual cost of the attack will be helpful on measuring the impact of this attack.
2. The hypothesis about the transfer asymmetry reflecting a “teacher-student” dynamic does not seem to have sufficient evidence. It would be much more interesting if we can actually verify the transfer asymmetry relationship with actual distilled models.
3. There is no ablation on the helper model used for style modification, which is a huge part of the attack.

**Limitations:**

yes

**Strengths And Weaknesses:**

## Soundness
- Strengths: The submission is highly technically sound, supporting its claims with both formal theoretical analysis and comprehensive empirical results. The theoretical framework provides a formal regret bound for the LinUCB policy that explicitly accounts for the non-linear model misspecification inherent in LLM evaluations. Empirically, the experiments are well-designed, testing the BITE framework across five diverse LLM judges (both open and closed source) and comparing it against strong baselines like prompt injection and optimization-based jailbreaks.

## Presentation
- Strengths: The paper is clearly written, well-structured, and easy to follow. The authors do an excellent job positioning their work; they clearly acknowledge prior literature that identifies stylistic biases as passive flaws, and articulate how their work differs by weaponizing these biases into an active attack surface. The use of visual aids, such as the overview diagram in Figure 1 and the "vulnerability fingerprint" heatmap in Figure 3, effectively communicates the attack dynamics and empirical findings.

## Significance
- Strengths: The paper addresses an extremely important and timely problem: the objective reliability of the LLM-as-a-judge paradigm. Because this paradigm is actively deployed in high-stakes pipelines like RLHF, chatbot leaderboards, and automated scientific peer review, the ability to artificially inflate scores has broad and severe implications. By exposing these vulnerabilities and demonstrating that standard style-control defenses are ineffective , this work will likely influence future research and necessitate the development of more attack-aware evaluation protocols.

## Originality
- Strengths: The core novelty lies in combining existing techniques in a highly creative way: framing the exploitation of known LLM stylistic biases as a contextual bandit problem. Using an adaptive LinUCB agent to optimize black-box prompts without gradient access introduces a fresh perspective on adversarial attacks against LLM evaluators.
- Weaknesses: The individual stylistic biases targeted by the framework (e.g., verbosity, markdown, emojis) are heavily based on existing literature. The originality is primarily in the automated exploitation of these biases rather than the discovery of the biases themselves.

---

> ### Author Rebuttal · Authors · 2026-03-25
>
> Thanks for acknowledging our soundness in formal theoretical analysis and comprehensive empirical results, clear presentation, addressing an important and timely problem and novelty.
>
> ---
>
> **S1 && W1: no new stylistic biases discovery**
>
> We agree that our work is not primarily about discovering new stylistic biases. Instead, our contribution is **orthogonal and complementary to prior bias-discovery work**. While existing studies identify what stylistic preferences LLM judges exhibit, our work asks how such biases can be systematically exploited through an adaptive black-box optimization framework. This distinction is important: bias discovery characterizes the vulnerability, whereas our method operationalizes it into a practical attack pipeline. Our framework is also flexible: newly discovered biases can be directly integrated into the action space, making the approach naturally extensible and able to benefit from future progress in bias discovery.
>
> ---
> **Q1: What is the actual attack cost?**
>
> Thank you for the suggestion. We agree that reporting the practical attack cost makes the budget constraint more interpretable. While we did not log exact per-sample billing statistics for every run in the paper, we extracted a representative estimate from a recent experiment. In this setup, we used Qwen3-235B as the judge model and Gemini-2.0-Flash-Lite as the helper model. Normalized to 100 samples, the estimated cost was approximately 1.77M tokens and \\$ 0.26 for the judge model, and 1.31M tokens and \\$ 0.19 for the helper model. These numbers should be viewed as approximate but representative estimates rather than exact per-sample accounting, since they are averaged over a mixed workload containing both pairwise and pointwise attack settings. We will add this practical cost estimate in the revision to make the real-world attack budget more explicit.
>
> ---
> **Q2: Explanation of transfer asymmetry**
>
> We propose this hypothesis as the most plausible explanation for the highly structured asymmetry we observe in Figure 4. While we do not have evidence, we believe these compelling asymmetric results are a key finding in themselves. They also strongly call for future work to validate this hypothesis and further explore the hidden data provenance relationships within the LLM ecosystem.
>
> ---
> **Q3: Ablation on the helper model**
>
> We thank the reviewer for pointing this out. We agree that the helper model is a meaningful design choice, and we have added an brief ablation with a fixed judge (Qwen3-235B) comparing Gemini-1.5-Flash-8B and GPT-4.1-Nano as helper model.
>
> | Setting | Helper Agent | Score Imp. |
> | :--- | :--- | :--- |
> | **Pointwise** | Gemini-1.5-Flash-8b | 2.69 |
> | | GPT-4.1-Nano | 2.87 |
> | **Pairwise** | Gemini-1.5-Flash-8b | 1.42 |
> | | GPT-4.1-Nano | 1.48 |
>
> The score improvement is consistent across both settings.
> This suggests that BITE does not depend on a specific helper model to succeed; rather, the helper mainly influences the quality of style-preserving rewrites at the margin. Thus, the core vulnerability lies in the judge’s stylistic bias and BITE’s adaptive exploration mechanism.

---

> > ### Author Rebuttal · Reviewer_1YdF · 2026-04-03
> >
> > My questions have been addressed. I will maintain my score.

---

### Decision · Program_Chairs · 2026-04-30

**Decision:**

Accept (regular)

**Comment:**

The paper studies how LLM judges can be manipulated to give positive scores through adversarially constructed stylistic changes to the input. 2/3 reviewers agree that the problem is well-motivated and proposed approach interesting. The remaining reviewer is unconvinced due to the threat model being unrealistic and the approach lacking as much novelty as the authors claim. In addition, there seem to be multiple issues with the math and proofs, as well as other minor errors.

Overall, I think this paper would be of sufficient interest to the ICML community to warrant publication; however, the authors should revise the paper with the clarifications and other fixes requested by the reviewers.

In addition to the revisions recommended by the reviewers, the authors should make the following formatting changes for a camera ready version:

- Please shorten your abstract. ICML strongly recommends that the abstract is a single paragraph, ideally between 4–6 sentences long. A longer abstract decreases the accessibility of your paper to readers.
- The font sizes in your figures also violate ICML's accessibility guidelines. The font size in figures should be no smaller than the font size of the caption of the figure. Figure 2 and Figure 5 in particular both need to be updated with larger font sizes, but you should consider a revision pass on all your figures.

Please see https://openreview.net/pdf?id=7g23tYAIDC and https://icml.cc/Conferences/2022/AccessiblePapersAndTalks for additional guidance.